# Morphology-based losses for weakly supervised segmentation of mammograms

**Mickael Tardy**[1,2]               MICKAEL.TARDY@EC-NANTES.FR
[1] *Ecole Centrale de Nantes, LS2N, UMR CNRS 6004, Nantes, France*
[2] *Hera-MI, SAS, Nantes, France*

**Diana Mateus**[1]                DIANA.MATEUS@EC-NANTES.FR

## Abstract

Segmentation is one of the most common tasks in medical imaging, but it often requires expensive ground truth for training. Weakly supervised methods cope with the lack of annotations, however, they often fall short compared to fully supervised ones. In this work, we propose to constrain the segmentation output with morphological operations, leading to an increase in the overall performance. In particular, we use top-hat and closing operations. We evaluate the method on high-resolution images from INBreast dataset and achieve an increase in $F_1$ of $\approx 0.14$ and in recall of $\approx 0.22$ compared to the training without morphology loss.

**Keywords:** Mammography, Weakly-supervised learning, Segmentation, Morphology

## 1. Introduction and Related Work

Breast cancer is one of the most spread diseases in the female population and is one of the leading causes of cancer death (Siegel et al., 2021). Recent clinical and technological advances allowed for early cancer detection leading to effective treatment and higher chances of recovery (Fisher et al., 2002). Emerging deep learning methods have boosted the field of computer-aided detection solutions (Geras et al., 2019). Lately self- and weakly supervised methods have drawn more attention (e.g., (Shen et al., 2020)), as they do not require explicit expert annotations in form of regions of interest of bounding boxes, which are often burdensome and expensive to collect. In this paper, we focus on the weakly supervised segmentation task for mammograms. Inspired by the work of (Kervadec et al., 2019) and based on the work of (Tardy and Mateus, 2021), we study loss functions based on morphological operators to constraint the segmentation output. That is, the size constraints (Kervadec et al., 2019) allow restricting the number of segmented pixels, but could lead to a substantial number of isolated regions. It is known that the top-hat operations are capable of removing isolated pixels. Therefore, we compare the weakly supervised segmentation mask with its post-processed image. In this way the network progressively integrates the post-processing step and makes less isolated false predictions. We also propose an improvement of (Tardy and Mateus, 2021) using a complementary loss term relying on the closing post-processing leading to further reduction of the number of isolated regions. To the best of our knowledge, we are the first to propose such constraints under a weakly supervised setting.

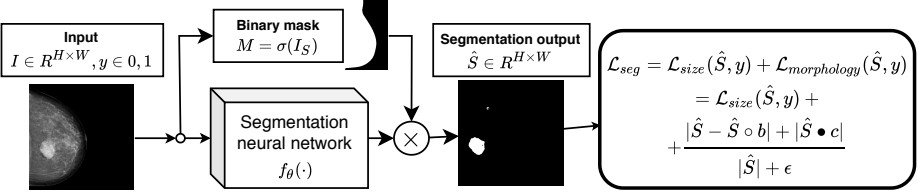

Figure 1: The overview of the proposed method with morphology loss $\mathcal{L}_{morphology}$ amongst the loss terms used for optimizing the neural network $f(\cdot)$.

## 2. Methods

Let $I_i$ be the input image of size $H \times W$, such that $I_i \in \mathbb{R}^{H \times W}$, and $y_i$ is its one-hot label, such that $y_i \in \{\mathcal{C}_k\}_{k=1}^K$. Let $f_\theta(\cdot)$ be a function with trainable weights $\theta$ and yielding a segmentation output, i.e., $\hat{S} = f_\theta(I)$, $\hat{S} \in \mathbb{R}^{H \times W}$. Having no pixel-wise ground truth, we propose to constrain the segmentation output as follows. First, as in (Kervadec et al., 2019) we use a $\mathcal{L}_s$ loss term setting a penalty for the regions outside expected size range. Second, as in (Tardy and Mateus, 2021), to reduce isolated pixels, we introduce the loss term based on the top-hat operation. With thresholding function $\sigma(\cdot)$, and having a structuring element $b$ and the opening operation $\circ$, the top-hat operation is $T = \sigma(\hat{S}) - \sigma(\hat{S}) \circ b$. Hence, the top-hat loss term is defined as $\mathcal{L}_{th} = \frac{|T|}{|\hat{S}|+\epsilon}$. Finally, we propose to use closing operation further reducing the number of isolated regions. Denoting the closing operator as $\bullet$ with the structuring element $c$, the closing operation $C$ is defined as $C = \sigma(\hat{S}) \bullet c$, and the proposed loss term is defined as: $\mathcal{L}_{cl} = \frac{|C|}{|\hat{S}|+\epsilon}$. Therefore, the complete morphology term is defined as follows:

$$\mathcal{L}_{morphology} = \mathcal{L}_{th} + \mathcal{L}_{cl} = \frac{|T| + |C|}{|\hat{S}| + \epsilon} \tag{1}$$

## 3. Experiments and Results

**Experimental setup**  We evaluate the proposed losses on a balanced subset of the IN-Breast dataset (Moreira et al., 2012) with five-fold cross-validation, having an equal proportion of normal and abnormal cases. In all experiments, the neural networks have been pre-trained in a self-supervised setting as in (Tardy and Mateus, 2021) using a subset of normal images and a range of synthesized artifacts. We resize the images to $2048 \times 2048$ and rescale the intensities to the range of $[0, 1]$. For the top-hat, we use the square element $b$ with side of $d_b = 3$. For the closing, we use the square element $c$ with side of $d_c = 50$.

**Results**  The metrics ($F_1$, Precision, Recall, True Positive Rate (TPR)) are reported in the Table 1. We also report Regions Count per image (RC/img), standing for the number of isolated regions on the segmentation mask. The best performances are achieved with the combination of the loss terms while each loss term alone (i.e., $\mathcal{L}_s$ and $\mathcal{L}_{morphology}$) yields the lowest results, which illustrate their complementarity.

Table 1: The results of the 5-fold cross-validation on the INBreast dataset

| Loss terms | $F_1$ | Precision | Recall | TPR | RC/img |
|---|---|---|---|---|---|
| $\mathcal{L}_s$ (Kervadec et al., 2019) | 25.96±7.08 | 36.19±5.19 | 28.74±7.47 | **100.0±0.0** | 46.95±10.43 |
| $\mathcal{L}_s + \mathcal{L}_{th}$ (Tardy and Mateus, 2021) | 38.18±6.01 | **46.14±3.94** | 42.13±7.24 | 96.97±3.03 | 9.96±3.08 |
| $\mathcal{L}_s + \mathcal{L}_{cl}$ | 39.47±5.22 | 44.89±5.81 | 48.35±6.93 | 93.94±2.14 | 4.91±0.78 |
| $\mathcal{L}_{morphology}$ | 28.26±3.18 | 22.45±3.41 | **63.77±2.80** | 96.97±2.14 | 5.88±0.52 |
| $\mathcal{L}_s + \mathcal{L}_{morphology}$ (ours) | **40.13±4.59** | 44.65±5.82 | 50.97±4.41 | 95.15±1.66 | **4.86±0.72** |

## 4. Discussion and Conclusion

In this work, we studied the effect of self-regulating loss terms based on morphological operations on the quality and the performances of the segmentation in a weakly supervised scenario. Our experiments have shown the benefits of such terms on segmentation output: it allowed increasing the overall $F_1$ score (from 25.96±7.08 to 40.13±4.59) while keeping comparable TPR and considerably reducing the number of isolated region predictions (from 46.95±10.43 to 4.86±0.72). More extensive study of the morphological operations, and in particular of the structuring elements properties, as well as loss weighing (see Equation (1)) could be done in future works. Also, as shown in (Tardy and Mateus, 2021), further improvements may be obtained with reconstruction objectives and synthesized imaging.

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
