# OpenReview forum: "Morphology-based losses for weakly supervised segmentation of mammograms"
_MIDL.io/2021/Conference/Short — MIDL 2021 Poster_

### Official Review · Reviewer_LQg4 · 2021-04-24

**Confidence:** 4
**Final Rating:** 3

**Summary:**

This paper proposed a morphology-based loss for regularization-based methods for weakly supervised segmentation.
The proposed method is an extension to their recently published journal paper ([1] Tardy and Mateus, 2021).
Experimental results show that the proposed method achieves better performance than previous methods.

[1] Mickael Tardy and Diana Mateus. Looking for abnormalities in mammograms with self-and weakly supervised reconstruction. IEEE Transactions on Medical Imaging, PP:1–1, jan 2021. ISSN 1558254X. doi: 10.1109/TMI.2021.3050040.

**Strengths:**

1. The paper is easy to understand and the proposed pipeline is clear.
2. The proposed loss term seems pluggable and easy to use.
3. The results show promising performance, compared to previous methods.

**Weaknesses:**

1. The novel design is only $\mathcal{L}_{cl}$, compared to their journal paper.
2. The intuition of why minimizing $\sigma (\hat{S}) \bullet c$ helps segmentation is not explained, which is different from $\sigma (\hat{S}) - \sigma (\hat{S}) \circ b$. The latter minimizes the difference of mask $\sigma (\hat{S})$ between before and after the openning operation $\circ$.

**Deanonymize Review:**

no

**Justification Of The Rating:**

This paper proposed an effective morphology-based regularization loss for weakly supervised segmentation, according to their experimental results. While their intuition of the loss design could be further explained.

**Paper Type:**

methodological development

**Special Issue:**

no

---

### Meta-Review · Area_Chair_Kvhg · 2021-05-06

**Recommendation:** Accept (Poster)
**Confidence:** 5

**Metareview:**

While unfortunately only one reviewer responded, the AC has studied the paper carefully and agrees with the reviewer. The paper fits well within the idea of presenting journal work at MIDL.

---

### Decision · Program_Chairs · 2021-05-11

Accept (Poster)